# Studies on the Proteinaceous Structure Present on the Surface of the *Saccharomyces cerevisiae* Spore Wall

**DOI:** 10.3390/jof9040392

**Published:** 2023-03-23

**Authors:** Yan Yang, Ganglong Yang, Zi-Jie Li, Yi-Shi Liu, Xiao-Dong Gao, Hideki Nakanishi

**Affiliations:** 1Key Laboratory of Carbohydrate Chemistry and Biotechnology, Ministry of Education, School of Biotechnology, Jiangnan University, Wuxi 214122, China; 2State Key Laboratory of Biochemical Engineering, Institute of Process Engineering, Chinese Academy of Sciences, Beijing 100190, China

**Keywords:** *Saccharomyces cerevisiae*, spore, cell wall, hydrophilin, RNA

## Abstract

The surface of the *Saccharomyces cerevisiae* spore wall exhibits a ridged appearance. The outermost layer of the spore wall is believed to be a dityrosine layer, which is primarily composed of a crosslinked dipeptide bisformyl dityrosine. The dityrosine layer is impervious to protease digestion; indeed, most of bisformyl dityrosine molecules remain in the spore after protease treatment. However, we find that the ridged structure is removed by protease treatment. Thus, a ridged structure is distinct from the dityrosine layer. By proteomic analysis of the spore wall-bound proteins, we found that hydrophilin proteins, including Sip18, its paralog Gre1, and Hsp12, are present in the spore wall. Mutant spores with defective hydrophilin genes exhibit functional and morphological defects in their spore wall, indicating that hydrophilin proteins are required for the proper organization of the ridged and proteinaceous structure. Previously, we found that RNA fragments were attached to the spore wall in a manner dependent on spore wall-bound proteins. Thus, the ridged structure also accommodates RNA fragments. Spore wall-bound RNA molecules function to protect spores from environmental stresses.

## 1. Introduction

Spores of the budding yeast *Saccharomyces cerevisiae* are dormant and stress-resistant forms of cells in which haploid nuclei generated via mitosis are enclosed [1,2]. Spores are generated in the cytosol of the mother cells in a manner such that the haploid nucleus generated via mitosis is enclosed by the newly synthesized spore plasma membrane and spore wall [2]. As a result, the mother cell becomes an ascus that includes four spores.

Spores have a spore wall that consists of multiple layers. It is believed that the outermost layer of the spore wall is primarily composed of ll-bisformyl dityrosine; this layer is called the dityrosine layer [2,3]. The dityrosine layer is most likely formed by crosslinking ll-bisformyl dityrosine molecules [4,5]. In support of this hypothesis, macromolecules containing dityrosine molecules are liberated by mild acid hydrolysis [5]. However, it is not clear how ll-bisformyl dityrosine molecules interact to form macromolecules [6,7]. Thus, the structure of the dityrosine layer remains elusive. Because ll-bisformy dityrosine is not digested by proteases or glycosidases, spores are resistant to attack by various digestive enzymes [5]. Additionally, the dityrosine layer confers resistance to some toxic chemicals, such as dimethyl ether, to spores [8]. While the dityrosine layer serves as a protective barrier for spores, it is not required to produce viable spores. For example, spores devoid of the dityrosine layer can be produced by the deletion of *DIT1*, which encodes an enzyme required for the synthesis of a precursor of ll-bisformyl dityrosine [3]. In *dit1*Δ spores, the chitosan layer is exposed at the spore surface. Electron microscopy analyses show that spores exhibit a ridged surface; however, the surface of *dit1*Δ spores is smooth [8].

Previously, we found that yeast spores were efficiently internalized in nonprofessional phagocytes (NPPs). Through the analysis of this phenomenon, we found that RNA fragments are noncovalently attached to the spore wall [9]. Spores are internalized by NPPs because RNA molecules are recognized by a phagocytic receptor RAGE (receptor for advanced glycation end-products). Spore wall-bound RNA molecules comprise various fragments derived from cytosolic RNA; thus, we speculate that mother cell RNA molecules that are not incorporated into spores are attached to the spore wall. The amount of RNA attached to the spore wall is significantly decreased by *dit1*Δ mutation. Of note, *dit1*Δ spores have a positively charged chitosan layer at the spore surface. Nevertheless, wild-type spores can hold more RNA than *dit1*Δ spores, suggesting that spores have the machinery to accommodate RNA in the spore wall [9]. Such machinery should involve protein because protease treatment causes a loss of spore wall-bound RNA. High-salt-washed spores were shown to adsorb higher levels of RNA molecules than *dit1*Δ spores, indicating that the RNA-binding machinery is covalently linked to the dityrosine layer [9]. Given that the spore wall is equipped with machinery to hold RNA, the RNA fragments may be beneficial for spores.

In this study, we performed analyses to verify that the RNA binding machinery is present on the spore wall. Although the nature of the RNA-binding machinery was not clarified by this analysis, our results suggest that ridges present on the spore surface were proteinaceous structures, including the RNA-binding machinery. The proteinaceous structure is most likely distinctive from the dityrosine layer.

## 2. Materials and Methods

### 2.1. Yeast Strains

The yeast strains and oligonucleotide primers used in this study are listed in Table 1 and Table 2, respectively. All strains were in the SK-1 strain background and sporulated with high efficiency. To disrupt *SIP18*, a DNA fragment was amplified by PCR, using pFA6a-His3MX6 [10] as the template and HXO681 and HXO682 as the primers. The PCR fragment was integrated into haploid cells AN117-16D and AN117-4B [11], and the resulting strains were mated to generate diploid *sip18*Δ cells. The *hsp12*Δ mutant was constructed in the same way, the primer pairs HXO694 and HXO695 were used to generate a knockout cassette, and pFA6a-His3MX6 was used as a template. The *sip18*Δ*gre1*Δ double mutant was constructed based on *sip18*Δ haploid cells; deletion of *GRE1* was performed with a knockout cassette generated with the primer pair HXO685 and HXO686. Gene pFA6a-TRP1 [10] was used as a template.

Yeast cells were cultured in YPAD (10 g/L yeast extract, 20 g/L peptone, 30 mg/L adenine, 20 g/L glucose). Agar (20 g/L) was added to prepare plates. Yeast cells were sporulated as previously described [12]. Briefly, yeast cells derived from a single colony were cultured overnight in 5 mL of YPAD liquid medium. A 0.1 mL aliquot of the culture was transferred into 5 mL of YPAcetate (10 g/L yeast extract, 20 g/L peptone, 20 g/L potassium acetate) and grown overnight. The cells were harvested by centrifugation, resuspended in 2% potassium acetate medium at a concentration of 3 × 10^7^ cells/mL, and cultured for 24 h.

### 2.2. Preparation of Spores

To release spores from asci, the asci were suspended in 5 mL of lyticase (β-glucanase) buffer (50 mM potassium phosphate buffer, pH 7.5, 1.4 M sorbitol), and 20 μL of lyticase (Sigma-Aldrich, St. Louis, MO, USA) stock solution (10,000 U/mL was dissolved in 500 μL of 50% glycerol) was added. After a 3 h incubation at 37 °C, the spores were washed twice with lyticase buffer. Then, the spores were resuspended in water and sonicated with an ultrasonic disruptor (Xinchen Biological Technology, Nanjing, China) to rupture the asci membrane. The sonication conditions were as follows: power—45%; duration—5 min, with cycles of 5 s on and 2 s off. Then, the spores were washed 3 times with 0.5% Tween-20 and 2 times with water. The number of spores was counted with a platelet counter.

Spores treated with high salt were prepared by incubating the spores in 0.6 M NaCl solution for 1 min, and the supernatant was removed. This process was repeated three times and then washed twice with water. To treat the spores with RNase A, 2 × 10^8^ spores suspended in 1 mL of water supplemented with 30 U/mL RNase A (TaKaRa, Tokyo, Japan) were incubated at 37 °C for 1 h. Spores were washed twice with water. To prepare protease-treated spores, 2 × 10^8^ spores suspended in 1 mL of water supplemented with 30 U/mL protease (Sigma-Aldrich) were incubated at 37 °C for 1 h and washed twice with water.

### 2.3. Extraction of RNA from the Spore Wall

A total of 2 × 10^8^ spores suspended in 500 μL of 0.6 M NaCl were vortexed at room temperature for 30 s. After centrifugation at 21,500× *g* for 10 min at 4 °C, RNA was purified from the supernatant with RNAiso plus (TaKaRa) according to the manufacturer’s instructions. The RNA concentration was measured using a NanoDrop 2000/2000c (Thermo Fisher Scientific, Cleveland, OH, USA).

### 2.4. RNA-Binding Assay to Salt-Washed Spores

A total of 2 × 10^8^ of salt-washed spores suspended in 1 mL of water were incubated with 40 μg/mL tRNA at 4 °C for 24 h with rotation. Then, the spores were centrifuged at 1000× g for 5 min at 4 °C and washed twice with water. To measure the amounts of tRNA bound to the spores, tRNA-bound spores were suspended in 200 μL of 0.6 M NaCl and vortexed at room temperature for 30 s. Then, the spores were centrifuged at 21,500× *g* for 5 min at 4 °C. RNA was purified from the supernatant with RNAiso plus. The RNA concentration was measured using a NanoDrop 2000/2000c.

### 2.5. Calcofluor White Staining

To stain spores with calcofluor white (CFW), 5 × 10^8^ spores or asci were suspended in 200 μL of 4% paraformaldehyde at room temperature for 10 min and washed twice with PBS. Then, the spores were mixed with 20 μL of 1 mg/mL CFW (Sigma-Aldrich). The mixture was incubated at 30 °C for 30 min and then washed three times with PBS. The cells were resuspended in 1 mL of water and then observed by fluorescence microscopy. Fluorescence quantification was measured with a microplate reader (Synergy H4, BioTek, Winooski, VT, USA) at an excitation wavelength of 355 nm and an emission wavelength of 440 nm.

### 2.6. Ether or β-Glucanase Sensitivity Assay

For the ether sensitivity assay, 10^7^ spores were suspended in 900 μL of distilled water. The suspension was mixed with 100 μL of diethyl ether and incubated for up to 16 min at 30 °C. Then, 100 μL of the spore suspension was removed at 8 min intervals and immediately mixed with 900 μL of water, and 100 μL of the sample was plated onto a YPAD plate. The plates were incubated at 30 °C for 36 h, and colony numbers were counted.

For the β-glucanase sensitivity assay, 10^7^ wild-type spores were suspended in water supplemented with 100 U lyticase and incubated at 37 °C for 2 h. Then, 5 μL of the spore suspension was mixed with 995 μL of water, and 50 μL of sample was plated onto a YPAD plate. The plates were incubated at 30 °C for 36 h, and colony numbers were counted.

### 2.7. HPLC Analysis of Dityrosine

Sample preparation and HPLC analysis were performed as described previously [6]. Two hundred microliters of 6 N HCl was added to 80 mg of spores suspended in 100 μL of water, and the suspensions were incubated at 95 °C for 5 h with an open lid. The dried hydrolysates were resuspended in 500 μL of water, vortexed at room temperature for 2 min, centrifuged at 21,500× *g* for 5 min, and filtered through 0.45 μM microfilters. The filtered samples were diluted 10 times to perform HPLC analysis. Dityrosine was synthesized by the oxidation of l-tyrosine, as previously described [6]. Briefly, 2 mL of Tris-HCl (0.3 M, pH 8.5), 1.5 mL of l-tyrosine (2 mg/mL), 0.1 mL of hydrogen peroxide (0.003%), and 0.5 mL of horseradish peroxidase (1 mg/mL; Sangon, Shanghai, China) were mixed and incubated at 20 °C for 1 h.

The samples were analyzed with a Discovery C18 column (150 mm × 4.6 mm inner diameter, 5 μm particles) (Sigma-Aldrich) using a Waters separation module e2695 HPLC system (Waters, Wexford, UK). Ten microliters of samples were loaded. The column was developed with a gradient of CH_3_CN in 0.01 M trifluoroacetic acid (0–50% CH_3_CN over 55 min). The flow rate was 1 mL/min. An excitation wavelength of 285 nm and an emission wavelength of 425 nm were used for detection. The quantification of dityrosine in the samples was performed based on peak area.

### 2.8. Light Microscopy

Light microscopy images were obtained using a Nikon C2 Eclipse Ti-E inverted microscope with a DS-Ri camera equipped with NIS-Element AR software (Nikon, Tokyo, Japan).

### 2.9. Field Emission Scanning Electron Microscopy (FESEM) Analysis of the Spore Wall

For the FESEM analysis of the spores, the spores were fixed in 5% glutaraldehyde (in PBS) overnight at 4 °C and then washed three times with 0.1 M phosphate buffer (pH 7.2). The samples were dehydrated using an ethanol series of 30% (20 min), 50% (10 min), 70% (10 min), 90% (10 min), 100% (10 min). Then, a critical point dryer (Leica, Wetzlar, Germany) was used to dry the samples. Images were acquired with FESEM instruments (Hitachi, Tokyo, Japan).

### 2.10. Mass Spectrometry (MS) Protein Analysis of the Spore Wall

One milligram of spores was washed with high salt (0.6 M NaCl) three times and then washed twice with water. After high salt wash, the spores were reduced with 250 μL of dithiothreitol (10 mM) at 37 °C for 1 h and alkylated with 250 μL iodoacetamide (15 mM) at room temperature for 30 min in the dark. After centrifugation at 1000× *g* for 2 min, the spores were suspended in 500 μL of NH_4_HCO_3_ (40 mM, pH 7.8), supplemented with 25 µg of sequencing-grade trypsin (Promega, Madison, WI, USA), and incubated for 16 h at 37 °C. After centrifugation at 21,500× *g* for 10 min at 4 °C, 350 μL of supernatant was collected and analyzed by liquid chromatography–tandem MS (LC–MS/MS). LC–MS/MS analysis was performed for 60 min on a Q-Exactive mass spectrometer (Thermo Fisher Scientific, Cleveland, OH, USA) that was coupled to an Easy nLC (Proxeon Biosystems, Odense, Denmark) from Shanghai Applied Protein Technology (Shanghai, China). MS/MS spectra were searched using MaxQuant software (version 1.5.3.17) against the UniProt proteome database (uniprot_Saccharomyces_cerevisiae _47897_20191018.fasta), and the label-free quantitation algorithm was performed for quantitative analysis. MaxQuant search was performed with a fragment ion mass tolerance of 0.01 Da and a parent ion tolerance of 20.0 PPM. The false discovery rates (FDRs) of protein groups and peptides were less than 0.01.

### 2.11. Determination of Spore Diameter

The longer diameters of spores were analyzed using ImageJ software [13]. The diameter of each spore was measured by drawing a straight line across the spore three times and calculating the average.

### 2.12. Statistics

All experiments were performed with at least three independent samples. Differences between the analyzed samples were considered significant at *p* < 0.05. Statistical significance was determined with two-tailed unpaired Student’s *t* test calculated with GraphPad Prism 8 software (GraphPad Software 8).

## 3. Results

### 3.1. RNA and Proteins Attached to the Spore Wall Have Protective Functions

Previously, we found that RNA fragments were attached to the spore wall [9]. Because the spore wall has a protective function, we speculated that spore wall-bound RNA may protect spores from environmental stresses. To assess this hypothesis, RNase-treated spores were subjected to ether (dimethyl ether) or β-glucanase treatments; these assays are often used to analyze the integrity of the spore wall [8,12]. As shown in Figure 1, spores treated with RNase exhibited sensitivity to these treatments (Figure 1A,B). The spore wall can accommodate RNA fragments in a manner dependent on spore wall-bound protein(s), suggesting that the presence of machinery that holds RNA in the spore wall [9]. Accordingly, protease-treated spores also exhibited ether and β-glucanase sensitivities (Figure 1A,B). These results show that RNA present on the spore wall is required to protect spores.

RNA bound to the spore wall was significantly decreased in the dityrosine layer-defective mutant *dit1*Δ [9]. Thus, the machinery to accommodate RNA is linked to the dityrosine layer. To analyze whether the dityrosine layer is compromised by protease or RNase treatments, a calcofluor white (CFW)-staining assay was performed. Because the dityrosine layer prevents the binding of CFW to chitosan in the spore wall, the integrity of the dityrosine layer can be analyzed by CFW staining [14]; for example, *dit1*Δ spores are stained by CFW (Figure 2A,B). However, the protease- or RNase-treated spores were not stained by CFW (Figure 2A,B). This result suggests that the dityrosine layer is not abrogated by these treatments. To further assess the effect of the treatments on the dityrosine layer, the levels of dityrosine in the spores were measured. In the protease- and RNase-treated spores, dityrosine levels were decreased by 21% and 15%, respectively (Figure 2C,D). Thus, dityrosine molecules were slightly removed by protease or RNase treatment. Nevertheless, most dityrosine molecules were impervious to protease and RNase treatment.

### 3.2. Proteinaceous Structure Is Present on the Surface of the Spore Wall

Next, we assessed whether the morphology of the spores was altered by the removal of RNA fragments or the RNA-binding machinery from the spore wall. To this end, we observed the RNase- or protease-treated spores by field emission scanning electron microscopy (FESEM). Spore images of FESEM are different from those of previous electron microscopy analyses [8], presumably because procedures to prepare the samples are different. Nevertheless, both images show that the surface of spores had a ridged appearance (Figure 3A,B). The morphology of spores was indistinguishable before and after RNase treatment (Figure 3C,D). However, we found that the surface of the spore wall became smooth after treatment with protease (Figure 3E,F). We counted spores with furrows as an index to evaluate the structural differences in the spore wall. As shown in Figure 3I, 75% of the protease-treated spores were devoid of furrows. Spores became smooth after protease treatment, possibly because the spores were swollen. However, longer diameters of spores were not altered before and after protease treatment (Figure 3J). Thus, the protease treatment most likely caused a removal of the ridged structure from the surface of the spore wall. These results suggest that a proteinaceous structure, which forms ridges, is present on the surface of the spore wall. The ridged structure was observed in the high-salt washed spores (Figure 3G,H), suggesting that the proteins are covalently linked to the spore wall. Of note is that cylindrical projections were found on the surface of the protease-treated spores (Figure 3F arrowhead). We speculate that the projections are scars from the interspore bridges that connect the spores in the asci [15].

### 3.3. Hydrophilins Are Concentrated in the Spore Wall

To identify proteins possibly attached to the spore wall surface, spores washed with high-salt solution were treated with trypsin, and peptides released from the spores were subjected to mass spectrometry analysis. Proteins identified by this analysis are listed in Appendix A. The analysis showed that a β-glucanase, Bgl2, exhibited the highest iBAQ value (Appendix A). Bgl2 is an abundant protein in the vegetative cell wall [16]; thus, a large amount of β-glucanase may be incorporated into the spore wall as well. Sip18 exhibited the second highest iBAQ value (Appendix A). Sip18 is a hydrophilin that is characterized by small, hydrophilic, and intrinsically disordered properties [17]. Of note, Gre1, which is a paralog of Sip18 [18], and another hydrophilin, Hsp12 [19], show higher iBAQ values in the list (third and fifth, respectively). The function of hydrophilins in spore or cell wall formation is unknown. However, the results suggest that the family members are included in the ridged and proteinaceous structure on the spore wall. Thus, we analyzed whether hydrophilins are required for RNA binding to the spore wall. However, the levels of RNA held to the spore wall were not reduced by the *sip18*Δ, *sip18*Δ*gre1*Δ, and *hsp12*Δ mutations (Figure 4A). Furthermore, we performed an in vitro tRNA-binding assay in which the salt-washed spores were incubated with tRNA, and the amount of tRNA bound to the spores was measured. The result showed that the ability to accommodate RNA was not reduced by the deletion of the hydrophilin genes (Figure 4B).

The above results show that the hydrophilins were not involved in the binding of RNA in the spore wall. However, by FESEM analysis, we found that spores with deep furrows were significantly decreased by the *hsp12*Δ mutation (Figure 5). The percentages of spores without furrow were also increased in *sip18*Δ and *sip18*Δ*gre1*Δ spores (Figure 5I). Next, we analyzed the sensitivity of the mutant spores to ether or β-glucanase treatments. The *hsp12*Δ spores exhibited neither ether nor β-glucanase sensitivities (Figure 6A,B). However, *sip18*Δ*gre1*Δ spores exhibited both ether and β-glucanase sensitivities (Figure 6A,B). The *sip18*Δ spores were sensitive to β-glucanase but not to ether treatment. Compared to the *sip18*Δ spores, the *sip18*Δ*gre1*Δ spores exhibited severe ether and β-glucanase sensitivities (Figure 6A,B). In addition, we measured the dityrosine levels in the mutant spores. In the *hsp12*Δ and *sip18*Δ*gre1*Δ spores, the levels of dityrosine in the spore wall were decreased by 14% and 15%, respectively (Figure 7A,B). Thus, although the dityrosine levels were slightly reduced, the results show that the dityrosine layer was not abrogated by the mutations. However, unlike protease-treated spores, the hydrophilin mutant spores were stained by CFW (Figure 7C,D). The levels of CFW staining in the *sip18*Δ*gre1*Δ spores were higher than those in the *sip18*Δ and *hsp12*Δ spores. Thus, the dityrosine layer may become CFW permeable due to the hydrophilin mutations. Overall, the results suggest that hydrophilin proteins are required for the proper organization of the spore wall.

## 4. Discussion

Here, we report that a ridged and proteinaceous structure is present on the surface of the *S. cerevisiae* spore wall. It was believed that the dityrosine layer was present at the outermost layer of the spore wall [3]. Dityrosine molecules are included in the ridged structure; however, most of the molecules remain in the spores after the removal of the ridged structure by protease treatment. Furthermore, the dityrosine layer was still present in the protease-treated spores, according to the biochemical and CFW staining assays. These results suggest that the ridged and proteinaceous structure is a structure distinct from the dityrosine layer.

Our results suggest that hydrophilins are included in the ridged structure. Previous reports have shown that hydrophilins are cytosolic proteins involved in tolerance to desiccation stress in various organisms, including *S. cerevisiae*; these proteins likely serve as membrane and protein stabilizers [20,21]. Furthermore, Hsp12 has been shown to be present in the cell wall of vegetative cells [20]. In this study, Sip18, Gre1, and Hsp12 are found as spore wall-bound proteins. Since hydrophilin mutant spores can hold RNA, these proteins are not involved in the RNA-binding machinery. Nevertheless, hydrophilin mutants exhibit spore wall defects. Thus, these proteins may serve as structural components to form the ridged structure on the spore wall. In addition, *hsp12*Δ spores show a smooth surface, which is reminiscent of protease-treated spores. However, unlike protease-treated spores, *hsp12*Δ spores can accommodate RNA fragments and do not show ether and β-glucanase sensitivities, indicating that the proteinaceous structure is present in mutant spores. Sip18 and Gre1 are paralogs, and defects in *sip18*Δ spores are exacerbated by an additional *gre1*Δ mutation, suggesting that these proteins have redundant functions in the spore wall. While our results suggest that the hydrophilins are required to form the spore wall properly, *hsp12*Δ and *sip18*Δ*gre1*Δ spores exhibit different morphological and pharmacological phonotypes. Thus, the function of Hsp12 may be distinct from that of Sip18 and Gre1. We found that hydrophilin mutants are stained by CFW. Given that the protease-treated spores were negative for CFW staining, the surface structure is dispensable for CFW-staining resistance. Thus, hydrophilins may also be involved in the proper organization of the dityrosine layer. It should be noted that we cannot rule out the possibility that hydrophilins are indirectly required to organize spore wall components. Further analyses are required to clarify the function of hydrophilins in spore wall formation.

Apart from hydrophilins, a variety of cytosolic proteins were detected from the analysis of the spore wall-bound proteins, although further studies are required to verify that these proteins are components of the proteinaceous structure. Because spores are formed inside the mother cell’s cytosol, we speculate that cytosolic proteins, which are not incorporated in spores, could be used as materials to construct the spore wall. Given that the proteinaceous structure persists on the spore wall through a high-salt wash, its components are covalently linked to the spore wall [9]. In the fission yeast *Schizosaccharomyces pombe*, the outermost layer of the spore wall is known to be composed of crosslinked Ips3 proteins [22]. Thus, crosslinked proteins may be prevalent in the yeast spore wall. The detailed mechanism for the crosslinking of Isp3 proteins in *S. pombe* spores remains unknown, and it would be intriguing to determine whether proteins are covalently linked in the nascent yeast spore wall.

The spore wall sequesters RNA fragments of cytosolic RNAs since RNA-binding machinery is present in the spore wall [9]. Given that proteinase treatment causes a loss of spore RNA fragments from the spore wall [9], the machinery is included in the ridged and proteinaceous structure. Spore wall-bound RNA molecules most likely function to protect spores. Extracellular polynucleotides found in bacterial biofilms are known to perform protective functions [23]. Thus, RNA molecules would be useful as a material that could protect cells. Our proteomic analysis showed that a variety of proteins that can bind to polynucleotides are included in the spore wall. Further studies of these proteins may clarify the machinery that accommodates RNA in the spore wall. Similar to *S. cerevisiae* spores, many fungal and bacterial spores are generated in the mother cell’s cytosol. Thus, it is tempting to ask whether RNA is attached to other spores.

## Figures and Tables

**Figure 1 jof-09-00392-f001:**
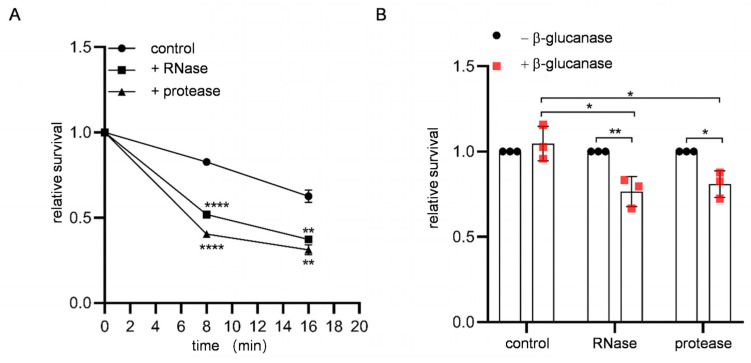
Stress sensitivities of spores treated with RNase or protease. (**A**) Spores treated with or without (control) RNase or protease were incubated with ether solution for the indicated times and plated on rich media, and colony numbers were counted. For each assay, the number of colonies obtained at 0 min (without ether treatment) was determined as 1, and the relative viability is shown. (**B**) Spores treated with or without (control) protease or RNase were treated with or without β-glucanase for 2 h at 37 °C and plated on rich media, and colony numbers were counted. Data are presented as the mean ± SEM. Statistical significance was determined by two-tailed unpaired Student’s *t*-tests. *n* = 3. *, *p* < 0.05; **, *p* < 0.01; ****, *p* < 0.0001.

**Figure 2 jof-09-00392-f002:**
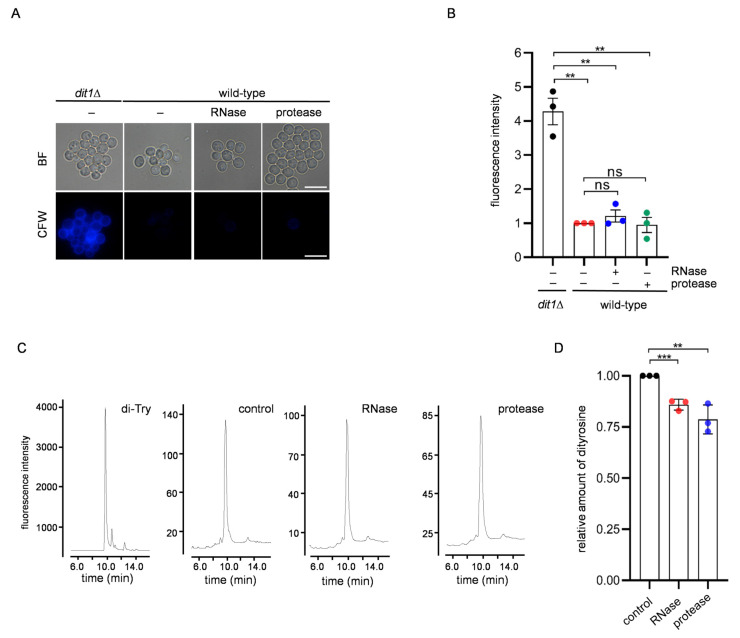
Detection of dityrosine in spores after RNase or protease treatment. (**A**) *dit1*Δ and wild-type spores treated with or without (−) RNase or protease were stained with CFW and observed under fluorescence (CFW) or bright field (BF) microscopy. Scale bar, 5 µm. (**B**) Quantification of the fluorescence intensities of CFW of cells described in (**A**). (**C**) Spores treated with or without (control) RNase or protease were hydrolyzed with 6 N HCl, and the lysates were subjected to HPLC analysis. Representative chromatograms are shown. Synthesized dityrosine (di-Tyr) was assayed as a control. (**D**) The relative amount of dityrosine detected by HPLC is described in (**C**). The dityrosine peak detected in the control sample was defined as 1. Data are presented as the mean ± SEM. Statistical significance was determined by two-tailed unpaired Student’s *t*-tests. *n* = 3. **, *p* < 0.01; ***, *p* < 0.001; ns, not significant (*p* ≥ 0.05).

**Figure 3 jof-09-00392-f003:**
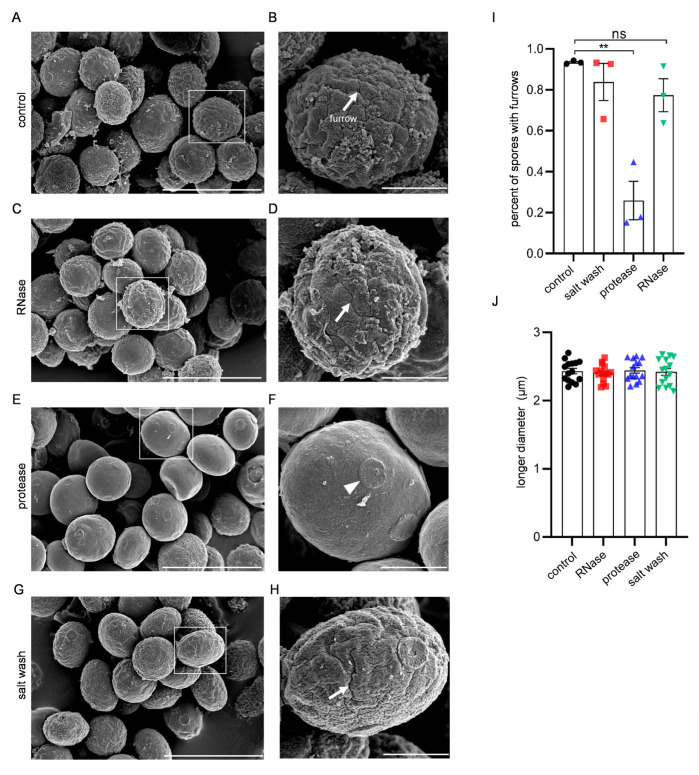
FESEM analysis of the surface morphology of spores. (**A**) Untreated control. (**C**,**E**,**G**) RNase-, protease-, or salt washed spores, respectively. (**B**,**D**,**F**,**H**) Close-up of the boxed areas in (**A**,**C**,**E**,**G**), respectively. (**A**,**C**,**E**,**G**) Scale bar, 5 µm. (**B**,**D**,**F**,**H**) Scale bar, 1 µm. Arrows in (**B**,**D**,**H**) indicate furrows. The arrowhead in (**F**) indicates a probable interspore bridge scar. (**I**) Quantification of spores with furrows. (**J**) Analysis of the longer diameters of spores. Data are presented as the mean ± SEM. Statistical significance was determined by two-tailed unpaired Student’s *t*-tests. *n* = 3 (**I**), *n* = 15 (**J**); **, *p* < 0.01; ns, not significant (*p* ≥ 0.05).

**Figure 4 jof-09-00392-f004:**
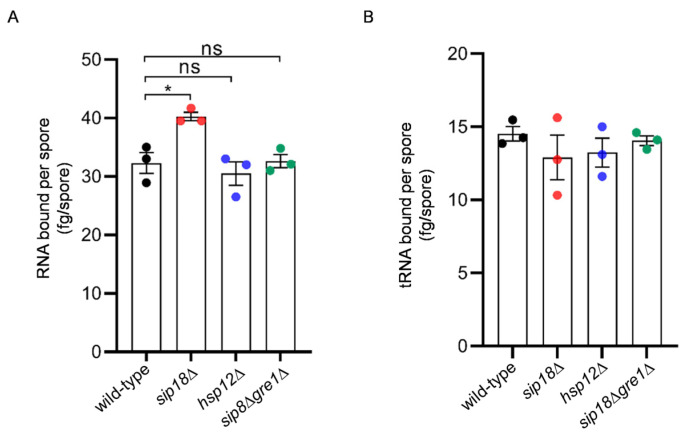
Analysis of the ability of hydrophilin mutant spores to hold RNA. (**A**) Amount of RNA eluted from 2 × 10^8^ wild-type, *sip18*Δ, *sip18*Δ*gre1*Δ, or *hsp12*Δ spores. (**B**) Amounts of tRNA bound to high-salt-washed wild-type *sip18*Δ, *sip18*Δ*gre1*Δ or *hsp12*Δ spores. A total of 2 × 10^8^ spores were incubated with 40 µg of tRNA. Data are presented as the mean ± SEM. Statistical significance was determined by two-tailed unpaired Student’s *t*-tests. *n* = 3. *, *p* < 0.05; ns, not significant (*p* ≥ 0.05).

**Figure 5 jof-09-00392-f005:**
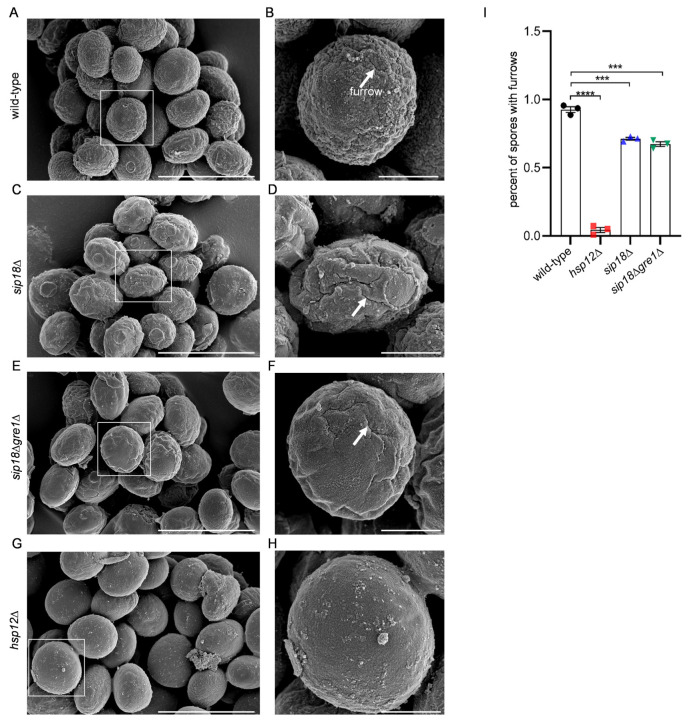
FESEM analysis of hydrophilin mutant spores. (**A**,**C**,**E**,**G**) Wild-type *sip18*Δ, *sip18*Δ*gre1*Δ, and *hsp12*Δ spores, respectively. (**B**,**D**,**F**,**H**) Close-up of the boxed areas in (**A**,**C**,**E**,**G**), respectively. (**A**,**C**,**E**,**G**) Scale bar, 5 µm. (**B**,**D**,**F**,**H**) Scale bar, 1 µm. Arrows indicate furrows. (**I**) Quantification of spores with furrows. Data are presented as the mean ± SEM. Statistical significance was determined by two-tailed unpaired Student’s *t*-tests. *n* = 3 (**I**). ***, *p* < 0.001; ****, *p* < 0.0001.

**Figure 6 jof-09-00392-f006:**
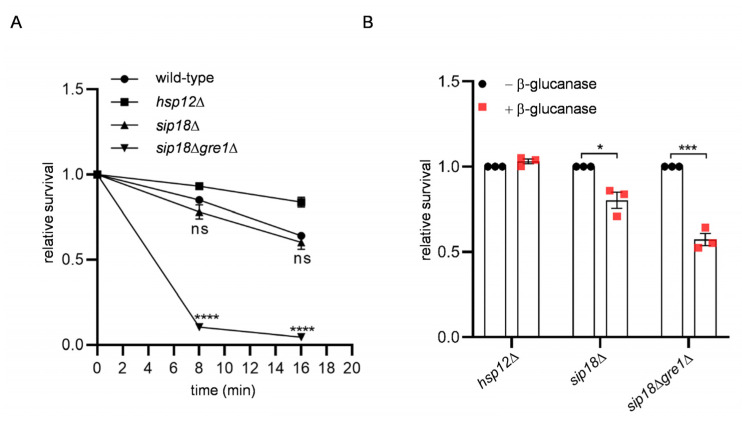
Stress sensitivity of hydrophilin mutant spores. (**A**) The indicated spores incubated with ether solution for the indicated times were plated on rich media, and colony numbers were counted. For each assay, the number of colonies obtained at 0 min (without ether treatment) were determined as 1, and the relative viability is shown. (**B**) The indicated spores treated with or without β-glucanase for 2 h at 37 °C were plated on rich media, and colony numbers were counted. Data are presented as the mean ± SEM. Statistical significance was determined by two-tailed unpaired Student’s *t*-tests. *n* = 3. *, *p* < 0.05; ***, *p* < 0.001; ****, *p* < 0.0001; ns, not significant (*p* ≥ 0.05).

**Figure 7 jof-09-00392-f007:**
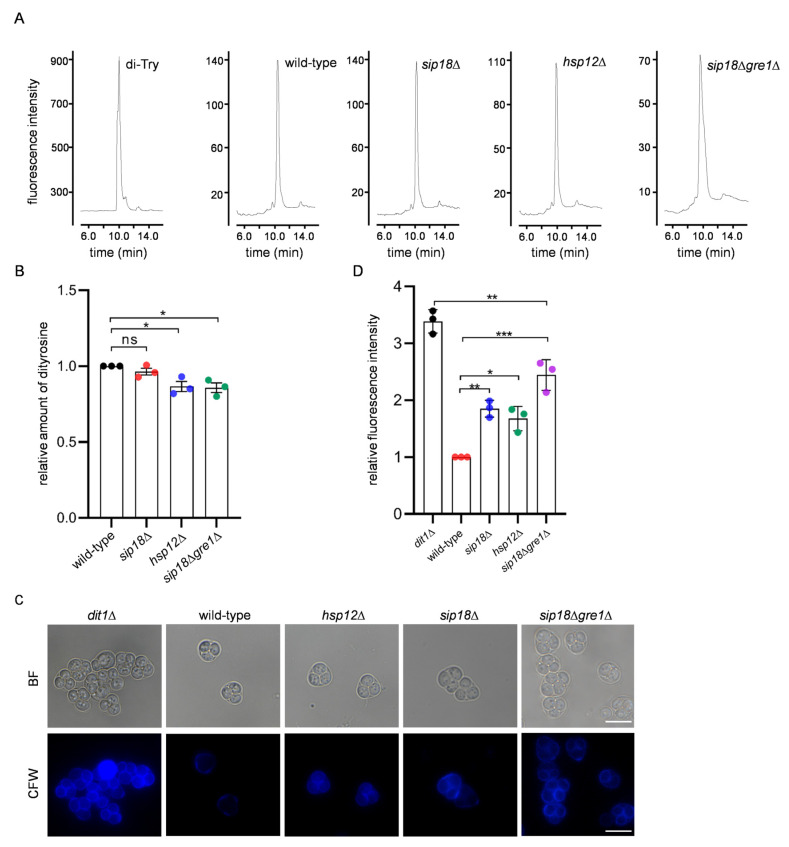
CFW staining assay of hydrophilin mutant spores. (**A**) Lysates of mutant spores were hydrolyzed with 6 N HCl and subjected to HPLC analysis. Synthesized dityrosine (di-Tyr) was assayed as a control. (**B**) The relative amount of dityrosine detected by HPLC described in (**A**). The dityrosine peak detected in the control sample was defined as 1. (**C**) The *dit1*Δ and wild-type *sip18*Δ, *sip18*Δ*gre1*Δ or *hsp12*Δ spores were stained with CFW and observed under fluorescence (CFW) or bright field (BF) microscopy. Scale bar, 5 µm. (**D**) Quantification of the fluorescence intensities of CFW of the indicated spores. Data are presented as the mean ± SEM. Statistical significance was determined by two-tailed unpaired Student’s *t*-tests. *n* = 3. *, *p* < 0.05; **, *p* < 0.01; ***, *p* < 0.001; ns, not significant (*p* ≥ 0.05).

**Table 1 jof-09-00392-t001:** Strains used in this study.

Strain	Genotype	Source
AN120	*MATα/MATa ARG4/arg4-NspI his3*Δ*SK/his3*Δ*SK ho::LYS2/ho::LYS2 leu2/leu2 lys2/lys2 RME1/rme1::LEU2 trp1::hisG/trp1::hisG ura3/ura3*	[11]
AN117-4B	*MATα ura3 leu2 trp1 his3*Δ*SK arg4-NspI lys2 ho::LYS2 rme1::LEU2*	[11]
AN117-16D	*MATa ura3 leu2 trp1 his3*Δ*SK lys2 ho::LYS2*	[11]
*dit1*Δ	*MATα/MATa ARG4/arg4-NspI his3*Δ*SK/his3*Δ*SK ho::LYS2/ho::LYS2 leu2/leu2 lys2/lys2 RME1/rme1::LEU2 trp1::hisG/trp1::hisG ura3/ura3 dit1*Δ*::his5^+^/dit1*Δ*::his5^+^*	[6]
*sip18*Δ	*MATα/MATa ARG4/arg4-NspI his3*Δ*SK/his3*Δ*SK ho::LYS2/ho::LYS2 leu2/leu2 lys2/lys2 RME1/rme1::LEU2 trp1::hisG/trp1::hisG ura3/ura3 sip18*Δ*::his5^+^/sip18*Δ*::his5^+^*	This study
*hsp12*Δ	*MATα/MATa ARG4/arg4-NspI his3*Δ*SK/his3*Δ*SK ho::LYS2/ho::LYS2 leu2/leu2 lys2/lys2 RME1/rme1::LEU2 trp1::hisG/trp1::hisG ura3/ura3 hsp12*Δ*::his5^+^/hsp12*Δ*::his5^+^*	This study
*sip18Δgre1*Δ	*MATα/MATa ARG4/arg4-NspI his3*Δ*SK/his3*Δ*SK ho::LYS2/ho::LYS2 leu2/leu2 lys2/lys2 RME1/rme1::LEU2 trp1::hisG/trp1::hisG ura3/ura3 sip18*Δ*::his5^+^/sip18*Δ*::his5^+^ gre1*Δ*::trp1/gre1*Δ*::trp1*	This study

**Table 2 jof-09-00392-t002:** Oligonucleotide primers used in this study.

**Name**	**Sequence**
HXO681	ATCTGTGCTCTTTACTTTAGTAGAAAGGTATATAAAAAAGTATAGTCAAGCGGATCCCCGGGTTAATTAA
HXO682	CATTAAAAAAATAAAAGGACTTGGTTAATTGCGCCCAAAAAAACGTAACAGAATTCGAGCTCGTTTAAAC
HXO685	CAGAGCCAAACACTACCGCATAAAAGCTAAGTACGAATAATAATTAAGAACGGATCCCCGGGTTAATTAA
HXO686	AGGGTGAGACCCGCACCTCAGGCATGTAATAGAAGCTTCGACCACCGCATGAATTCGAGCTCGTTTAAAC
HXO694	TTCGATAATCTCAAACAAACAACTCAAAACAAAAAAAACTAAATACAACACGGATCCCCGGGTTAATTAA
HXO695	TCACACATCATAAAGAAAAAACCATGTAACTACAAAGAGTTCCGAAAGATGAATTCGAGCTCGTTTAAAC

## Data Availability

The data supporting the findings of this study are available within the Article and its Appendix A. Further relevant data were available from corresponding authors upon reasonable request.

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
