# Peer review of "Studies on the Proteinaceous Structure Present on the Surface of the Saccharomyces cerevisiae Spore Wall"

_jof, 2023, doi:10.3390/jof9040392_

Round 1

Reviewer 1 Report

This manuscript from Yang et al. describes the presence of the proteinaceous structure on the surface of the yeast spore wall. The outermost layer of the yeast spore wall is believed to be consist mainly of dityrosine. However, how the dityrosine molecules interact to form macromolecules remains unknown. In this study, the authors identified a number of spore wall proteins, some of which are involved in the surface structure of spore. This is a very nice piece of work. The results are mostly solid and well presented. In addition, the manuscript is well written and will be interesting not only to yeast researchers but also general readers. Therefore, I recommend this manuscript for publication in Journal of Fungi without any addition experiments. 

I have several comments as listed below.  

I think that the surface structure of wild-type spore observed in this study is somewhat different from that reported in previous study from Couccio et al Microbiology 2004 (DOI 10.1099/mic.0.27253-0). I think that the authors should comment about this. 

Could the hydrophilins identified in this study be involved in the formation, but not the structure, of the dityrosine layer?

Did you observe the localization of the hydrophilins by fluorescence microscopy?

It would be easier to see the graph if it were larger.

The authors should describe the protease treatment in Materials and Methods section.

Line 81: "YPAcatate" should be " YPAcetate ".

Line 421: page 1643-1655. 

Ref22: First names are listed. They should be family names. 

Author Response

I would like to thank for valuable comments. Our responses are follows (line numbers are for the document with track changes).

  1. I think that the surface structure of wild-type spore observed in this study is somewhat different from that reported in previous study from Couccio et al Microbiology 2004 (DOI 10.1099/mic.0.27253-0). I think that the authors should comment about this. 

Response:

     The images are different may be because of difference in sample preparation procedures. Nevertheless, both images show that the surface of spores had a ridged appearance. This description has been added in the revised manuscript (line 252-255).

  1. Could the hydrophilins identified in this study be involved in the formation, but not the structure, of the dityrosine layer?

Response:

     Since hydrophilin mutans become CFW staining positive, the proteins may be also involved in the formation of the dityrosine layer (line 319-324, 378-381). It is notable that, we cannot rule out the possibility that hydrophilins are indirectly required to organize spore wall components. This possibility is now added in the revised manuscript (line 381-383).

  1. Did you observe the localization of the hydrophilins by fluorescence microscopy?

Response:

     We tried to find localization of Hsp12 and Sip18. However, their localization to the spore wall was not clearly defied because their signals were not strong in spores. Thus, more detailed analysis of these and other spore wall-localized proteins will be continued, and they will be published in a different study.

  1. It would be easier to see the graph if it were larger.

Response:   

     We have modified size of all figures.

  1. The authors should describe the protease treatment in Materials and Methods section.

Response:

     Method for proteinase treatment has been described in the section 2.3 Preparation of spores (line 113-115).

  1. Line 81: "YPAcatate" should be " YPAcetate ".
  2. Line 421: page 1643-1655. 
  3. Ref22: First names are listed. They should be family names.

Response for 6-8:

     We have corrected these mistakes.

Reviewer 2 Report

The manuscript by Yang et al presents a set of experimental evidences oriented to explain the proteinaceous structure of the yeast spore wall. The experiments are performed based in the assumption that RNA and proteins form structural part of the yeast spore. However, while the proteinaceous layer is expected to be part of the spore cell wall, the presence of RNA as structural component of this structure is not clear for this reviewer despite the fact that RNA could decorate the yeast cell wall.

In my opinion the authors do not demonstrated sufficiently the specific function of RNA in the biology of spores, but the importance of hydrophilins in the spore biology seemed relevant. However, the manuscript is rather descriptive and it do not addressed the reason for what hydrophilins are relevant, ignoring the potential role of them in the construction of the cell wall spore, therefore the results presented seemed insufficient to explain such a role.

Independently of the general opinion presented above the manuscript has some technical questions that need to be addressed before publication:

-The results on the role of RNA in the structure of the spore cell walls are inconclusive and therefore irrelevant for this manuscript. Here they only divert the focus from other more important results. I have not doubts on their presence in the spores, but their potential role on spore biology is only associated with the effect of the RNAse treatment, which could affect other structures of the spore since the authors ignore the potential contamination of RNAse A with other enzymatic activities, by example proteases.  Takarabio does not specify that its RNAse A is free of proteases.

- How the authors can be sure that the proteins identified in the MS assay really form part of the spore cell wall? The authors purified the spores, but they could be partially contaminated with protein rests from the “mother” cell despite extensive washing. In addition, it is possible that spore could be damage by the treatments with the DTT, iodocatemide and ammonium carbonate used, liberating intracellular proteins to the media. This does not mean that they could no be there, but the authors should demonstrate that these proteins are really there.

-Why the authors have chosen Sip18, Gre1 and Hps12 as target for further studies ignoring, by example, Bgl2? I believe that characterizing the effect of mutations in some of other major MS hits would really improve the conclusion of this work.

-The authors need more conclusive evidence on the changes produced by the mutations used. A simple scanning electronic microscopy is not enough to reveal changes in the structure of spore walls and conventional transmission EM is necessary to reveal the structure of the spore walls in the different strains. Note that the effect of Hsp12 absence is very strong on spore morphology, but null in spore resistance, unlinking scanning images to spore biology.

The effect of the different mutation on di-Tyr is really weak, and the florescence analysis performed with calcofluor is not conclusive enough due to differences in the accessibility of CW to chitosan layer, which can be variable depending in the experiment. Panels 2C and 7A are beautiful but absolutely irrelevant, di-Tyr is always di-Tyr, and quantitative data are already presented numerically in other panels.

-The evidence for a role of hydrophilins in spore biology is conclusive, but the authors have not demonstrated that it is linked to any role in the structure of spore cell wall. Such a role could be linked to a more general effect in the sporulation process, a possibility that has not been contemplated. This is especially clear in the case of Hsp12, which I suspect has a general role in the biology of yeast that could affect the sporulation and spores in many ways, not necessarily linked to its presence in the structure of spore walls. Note that its absence has no effect on glucanase or ether treatment, which is surprising observing the smooth aspect of its spores.

-Be careful with images legends, there a several mistakes along them.

-The manuscript will benefit from an editorial English revision.

            In conclusion, I found some of the results presented really interested, but rather preliminary.

            To be positive, I really believe that the involvement of hydrophilins in spore biology is a relevant scientific conclusion that deserves consideration, but it has to be accompanied by a more thorough analysis of the role of these proteins on spore biology and structure before to be published.

Author Response

I would like to thank for valuable comments. Our responses are follows (line numbers are for the document with track changes).

  1. The results on the role of RNA in the structure of the spore cell walls are inconclusive and therefore irrelevant for this manuscript. Here they only divert the focus from other more important results. I have not doubts on their presence in the spores, but their potential role on spore biology is only associated with the effect of the RNAse treatment, which could affect other structures of the spore since the authors ignore the potential contamination of RNAse A with other enzymatic activities, by example proteases.  Takarabio does not specify that its RNAse A is free of proteases.

Response:

     We cannot rule out the possibility that proteinase is contaminated in the RNase. However, even if proteinase was contaminated, the effect should be negligible since morphology of the spore wall was not altered by RNase treatment (please refer Figure 3 B, D, and F).

     This study reports that ridged and proteinaceous structure is present on the spore wall. Spore wall-bound RNA was found in a previous our study. But, as described in the manuscript, RNA molecules were found to be attached to the proteinaceous structure and they were likely involved in protection of spores. Thus, we believe that the findings regarding RNA can be incorporated in this paper.

  1. How the authors can be sure that the proteins identified in the MS assay really form part of the spore cell wall? The authors purified the spores, but they could be partially contaminated with protein rests from the “mother” cell despite extensive washing. In addition, it is possible that spore could be damage by the treatments with the DTT, iodocatemide and ammonium carbonate used, liberating intracellular proteins to the media. This does not mean that they could no be there, but the authors should demonstrate that these proteins are really there.

Response:

     As the reviewer mentions, the MS result is not perfectly conclusive. Thus, in this study, we present the data as a list of proteins possibly attached to the spore wall. However, the result is still meaningful since we found that hydrophilin are required for making the spore wall. It should be noted that hydrophilins may not be structural components of the spore wall and they are indirectly involved in spore wall formation. Thus, we have added this possibility in discussion (line 381-384). Furthermore, the manuscript has been modified, regarding this comment (line 21-22, 280, 367-368).

     We tried to find localizations of Hsp12 and Sip18. However, their localization to the spore wall was not clearly defied because their signals were not strong in spores. Thus, more detailed analysis of these and other spore wall-localized proteins will be continued, and they will be published in a different study.

  1. Why the authors have chosen Sip18, Gre1 and Hps12 as target for further studies ignoring, by example, Bgl2? I believe that characterizing the effect of mutations in some of other major MS hits would really improve the conclusion of this work.

Response:

     By the MS analysis, we found that several hydrophilin members are attached to the spore wall. It is not reported that hydrophilins are required for spore/cell wall formation. However, the result suggest that the family members are involved in the formation of the ridged and proteinaceous structure. Thus, we focused on these proteins. This explanation has been included in the text (line 287-291).

  1. The authors need more conclusive evidence on the changes produced by the mutations used. A simple scanning electronic microscopy is not enough to reveal changes in the structure of spore walls and conventional transmission EM is necessary to reveal the structure of the spore walls in the different strains. Note that the effect of Hsp12 absence is very strong on spore morphology, but null in spore resistance, unlinking scanning images to spore biology. The effect of the different mutation on di-Tyr is really weak, and the florescence analysis performed with calcofluor is not conclusive enough due to differences in the accessibility of CW to chitosan layer, which can be variable depending in the experiment. Panels 2C and 7A are beautiful but absolutely irrelevant, di-Tyr is always di-Tyr, and quantitative data are already presented numerically in other panels.

Response:

     We understand that the EM images are “not enough to reveal changes in the structure of spore walls”.  Therefore, we have provided further evidences, including CFW staining, ether sensitivity, and glucanase assays, to support the hypothesis that the hydrophilin mutations cause defects in the spore wall. It is true that the data are not conclusive to identify the defect(s). However, they still show that hydrophilin mutants are defective in the spore wall, which is the point we want to argue in this study.

     In Fig 2 and 7, di-Tyr assay results are presented primarily because we want to show that the dityrosine layer is not abrogated by the mutations or treatments (line 233-236, 316-320).

  1. The evidence for a role of hydrophilins in spore biology is conclusive, but the authors have not demonstrated that it is linked to any role in the structure of spore cell wall. Such a role could be linked to a more general effect in the sporulation process, a possibility that has not been contemplated. This is especially clear in the case of Hsp12, which I suspect has a general role in the biology of yeast that could affect the sporulation and spores in many ways, not necessarily linked to its presence in the structure of spore walls. Note that its absence has no effect on glucanase or ether treatment, which is surprising observing the smooth aspect of its spores.

Response:

     The reviewer’s major criticism is that hydrophilins may not be directly required for spore wall organization. Thus, in the revised manuscript, this possibility has been included in discussion (line 381-384). Furthermore, Abstract has been revised based on the criticism (line 21-22).

     The reason for why hsp12∆ spores do not exhibit ether and glucanase sensitivities remains unclear. However, morphological and CFW staining abnormalities still suggest that the mutant spores have a defect in the spore wall.

  1. Be careful with images legends, there a several mistakes along them.

Response:

     We have checked Figures and their legends and revised them. Particularly, scale bars have been added in Figure 5 A, C, E, G, and Figure 7C.

  1. The manuscript will benefit from an editorial English revision.

Response:

     English proofreading has been already performed before submission. However, we will follow the editor’s advice, if further proofreading is required for the revised manuscript.

Reviewer 3 Report

Studies on the proteinaceous structure present on the surface of the Saccharomyces cerevisiae spore wall presents the reader with a detailed analysis of the spore wall.

Overview:

  While the authors present numerous experimental approaches to ascertain the role of the dityr layer and elucidate the structural role of this layer many questions remain unanswered. This is not due to the authors inability to determine the role or the structure but more so due to the complexity of the layers and function of the spore wall in Saccharomyces cerevisiae.

Abstract – the authors make clear the importance of the crosslinked dipeptide, bisformyl dityrosine, however the significance of the RNA found in the spore wall is not described in sufficient detail.

Introduction – The authors do a nice job of giving rationale for spore formation and in defining the role of the protein components of the spore wall. What is lacking is the role/function of the RNA found associated with the spore wall. More introductory information to assist the readers in understanding why RNA would be associated with the spore wall is needed. This would strengthen not only the introduction but also their reasoning for the RNA in the discussion section. This phenomenon is intriguing and should be explored in the introduction especially since it is a distinct part of the discussion. Define what authors mean by proteinaceous machinery.

Methods – section complete

Results – a few issues with English usage in this section – ie line 192 – decollated – reviewer believes – incorrect terminology used here. Also, authors stress using mutant in their assay that RNA sequestered at spore wall is protective in nature. Their conclusions would be strengthened if perhaps they treated the spores with RNAse and then tested for the either and B-gluconase sensitivities to ensure their conclusions of protection are correct.

Section 3.2 – authors should define proteinaceous machinery – how does this align with RNA present in spore wall.

Discussion while much more work is needed to define the structure of the spore wall this article is a good first step although clarity of RNA and hydrophilins could be addressed here to strengthen this manuscript.

Author Response

I would like to thank for valuable comments. Our responses are follows (line numbers are for the document with track changes).

  1. Abstract – the authors make clear the importance of the crosslinked dipeptide, bisformyl dityrosine, however the significance of the RNA found in the spore wall is not described in sufficient detail.

Response:

     Based on the reviewer’s suggestion, we have modified description of RNA in Abstract (line 22-25). 

  1. Introduction – The authors do a nice job of giving rationale for spore formation and in defining the role of the protein components of the spore wall. What is lacking is the role/function of the RNA found associated with the spore wall. More introductory information to assist the readers in understanding why RNA would be associated with the spore wall is needed. This would strengthen not only the introduction but also their reasoning for the RNA in the discussion section. This phenomenon is intriguing and should be explored in the introduction especially since it is a distinct part of the discussion. Define what authors mean by proteinaceous machinery.

Response:

     Based on the reviewer’s suggestion, we have added description about spore wall-bound RNA in Introduction (line 52-56, 66-68).

  1. Results – a few issues with English usage in this section – ie line 192 – decollated – reviewer believes – incorrect terminology used here.

Response:

     We have revised the text according to the reviewer’s comment (line 204,397).

  1. Also, authors stress using mutant in their assay that RNA sequestered at spore wall is protective in nature. Their conclusions would be strengthened if perhaps they treated the spores with RNAse and then tested for the either and B-gluconase sensitivities to ensure their conclusions of protection are correct.

Response:

      We performed RNase/glucanase, and RNase/ether sensitivity assays (line 206-210). The results are shown in Figure 1A and B.

  1. Section 3.2 – authors should define proteinaceous machinery – how does this align with RNA present in spore wall.

Response:

     The original manuscript may be confusing because “proteinaceous” was used for both “RNA-binding machinery” and “ridged structure”. In the revised manuscript, proteinaceous has been only used for the “ridged structure”. Regarding this comment, we have described that the RNA-binding machinery is included in the ridged and proteinaceous structure (line 397-401). But further studies are required for clarification of the machinery. Hopefully, further analysis of the spore wall-binding proteins could provide insight into the machinery (line 406-407). These explanations are now included in Discussion.

  1. Discussion while much more work is needed to define the structure of the spore wall this article is a good first step although clarity of RNA and hydrophilins could be addressed here to strengthen this manuscript.

Response:

     Regarding hydrophilins, we described that hydrophilins are not involved in the RNA-binding machinery. Nevertheless, hydrophilins may be used as structural components to form the ridged structure on the spore wall (line 365-368). Hydrophilins may be indirectly required to organize spore wall components. This possibility is also described in the text (line 381-383)

     Regarding the RNA (-binding machinery) we modified the discussion as described in the answer of the previous comment (line 397-401).

Round 2

Reviewer 2 Report

The authors have addressed point by point my comments and, accordingly, they have modified the text to avoid conclusions not fully supported by the results presented.

However they have not performed any additional experiment and therefore my main concerns about the manuscript still remain, specially since the authors agrees in the lack of definition of some of their results (see by example first sentences of answers 2 and 4).

I still think that this manuscript does not address properly the biological role of hydrophilins in spore formation, therefore the results presented here are interested but descriptive and therefore rather preliminary and will require additional experimental evidence before publication.